# Molecular Dynamics Investigation of Spreading Performance of Physiological Saline on Surface

**DOI:** 10.3390/ma15113925

**Published:** 2022-05-31

**Authors:** Jianhua Pan, Xiao Wang

**Affiliations:** College of Physical Education, Chongqing University, Chongqing 400044, China; panjhua@cqu.edu.cn

**Keywords:** physiological saline, infiltration, molecular dynamics simulation, electric field

## Abstract

Physiological saline is an indispensable element for maintaining the functions of life. The spreading performance of physiological saline droplets on the surface of graphene under different NaCl concentrations and electric field intensities was studied in the present work. The results show that the increase in NaCl concentration reduces the displacement vector value of molecules in droplets. In addition, NaCl is easy to aggregate on the surface of graphene. The increase in NaCl concentration makes it more difficult for droplets to penetrate the surface of graphene, and the penetration angle of droplets increases with the rise in NaCl concentration. With the increase in electric field intensity, the wetting effect of droplets is more obvious. The greater the electric field intensity is, the smaller the penetration angle is, which is mainly due to the polarity of water molecules. This study has reference significance for the study of body fluid volatilization on the human surface.

## 1. Introduction

Sweat is an indispensable substance in human body and has the functions of maintaining body temperature, accumulating energy, and maintaining cell activity and human function [1]. However, although the skin cell tissue can effectively reduce the loss of ions in the body, it still cannot prevent ions from penetrating the skin surface with water under certain conditions [2,3,4,5]. Especially after a large amount of exercise, the particles in the body fluid will be lost, and the fast-flowing blood will transport the heat of other parts of the body to the skin surface within a short time, so as to accelerate the loss of water and ions in the body [6,7,8,9,10].

The evaporation and diffusion of nano-droplets is a relatively novel research method that has many potential applications in the fields of DNA chip manufacturing, medicine, inkjet printing, surface coating, and self-assembly based nanostructure formation [11,12]. Similarly, such ideas and research methods can also be applied to the evaporation of sweat on the surface of human skin. The inherent charge of nano-droplets competes between the pulse arrangement of dipole fluid molecules and the rearrangement induced by external electric fields, which leads to a change in the surface tension and finally a change in the droplet contact angle [13,14]. It has been found that the expansion behavior and deformation dynamics of nano-droplets can be significantly changed by adding an appropriate electric field. Although it has been found that the addition of salty ions and electric field improves the rate of evaporation process, its basic mechanism still needs to be further understood [15].

The simulation of molecular dynamics (MD) is widely used in materials, machinery, medicine, aerospace, and other fields to study the physical and chemical properties of microscopic substances [16,17,18]. MD simulation simulates the real behavior of nature by identifying each atom and tracking its trajectory in time through the basic laws of classical mechanics [19,20,21,22]. It is also used in human and medical research. Chanmugam et al. [23] used a three-dimensional computational model to quantitatively analyze the thermal properties of breast cancer. Only the effects of tumor size and depth on the thermal contrast (maximum temperature difference) under steady-state conditions were analyzed [24]. The effects of the cooling load on deep tumors, such as the effects of cooling time and temperature on the thermal contrast, have not been deeply analyzed in the literature. Chaban V evaluated the applicability of pyridine and pyrrolidine RTIL for solvent-assisted stripping of graphene using all-atom pairwise potential MD simulation. The results show that the interaction between the aromatic ring of BPY+ and the aromatic system of graphene does not provide significant advantages in stripping applications [25]. Guilherme Colherinhas applied density functional theory to study the effect of neutral ion clusters on a tGQD surface. We conclude that Homo and LUMO of GQD are very sensitive to the presence of ions and their distance from the surface of GQD [26].

As a large number of fluid substances are distributed on the skin’s surface, human sweat has an important heat-transfer function on this surface. This is also an important factor affecting athletes’ long-term, high-load exercise under high-temperature conditions [25,26,27]. However, although many researchers have conducted a lot of research on the body fluid loss of athletes, the infiltration laws of physiological saline on the skin surface have still been ignored. Therefore, few people have studied the infiltration of body fluid on the surface at the micro scale with the MD method. In order to better reveal the infiltration effect of liquid on the skin surface from the micro scale and explore the diffusion performance of physiological saline under different concentrations and electric field intensities, a three-dimensional mixing model of a physiological saline and graphene surface was established through molecular dynamics simulation [27,28,29,30]. In order to eliminate the error caused by skin surface roughness and more intuitively observe the impact of NaCl concentration and electric field intensity in physiological saline on its infiltration, the skin tissue structure was simplified to the graphene surface to simplify the relevant model.

## 2. Methodology

### 2.1. Creation of Simulation Model

The molecular simulation software Materials Studio (MS) was employed to create the simulation model. Firstly, the disordered simulation structure in Figure 1b was established by using the Amorphous Cell module, and then the H_2_O and NaCl molecules outside the black circle were deleted to obtain the initial model shown in Figure 1b. The parameters of the simulation system in the X, Y, and Z directions are 119 Å × 29 Å × 100 Å. The total number of atoms in each simulation system is 7910.

### 2.2. Computational Details

The well-known software of Large-scale Atomic/Molecular Massively Parallel Simulator (LAMMPS) is utilized to undertake all the calculation work [31]. The interaction between atoms is described by the POLS-AA force field [32]. The Lennard Jones potential is used in these simulations, and the cut-off distance and timestep of interaction are 12 Å and 1 fs, respectively. In addition, the Ewald sum method is employed for the calculation of long-range Coulomb interaction. Furthermore, in order to simulate the real physical environment, it is essential to set periodic boundary conditions in the X, Y, and Z dimensions. The temperature of the simulation system is set at room temperature and controlled by the nose Hoover thermostat algorithm. Since the energy of the whole system cannot be minimized at the beginning of modeling, all systems are optimized for energy minimization. A 2 nanosecond NVT (Particle number N, Volume V and temperature T of the system) constant temperature relaxation was carried out to reduce the system energy before each simulation. The electric field intensity was set to 0.02 V/Å, 0.04 V/Å, 0.06 V/Å, and 0.08 V/Å, and the different concentrations of NaCl in the simulation systems is shown in Table 1.

## 3. Results and Discussion

### 3.1. Displacement Distribution of Physiological Saline

Figure 2 shows the displacement distribution of physiological saline infiltration with different NaCl concentrations. In this process, only the influence of NaCl concentration was considered, and no electric field was applied. It is obvious that in the initial stage of calculation, the displacement distribution of molecules in each system was relatively small. With the increase in simulation time, the Brownian motion of molecules in each system became obvious and the displacement increased gradually. Apparently, without NaCl addition, the displacement distribution value of pure water at 0 ns was greater than that of any other systems, as shown in Figure 1a. On the contrary, at 0 ns, the displacement distribution of physiological saline at concentrations of 3% and 13.2% NaCl was relatively small, but with the increase in simulation time, the displacement distribution in the two systems gradually remained consistent with other systems due to the increase in the probability of intermolecular contact, as shown in Figure 2b,d. Among them, when the NaCl concentration was 3%, the displacement distribution value at 2 ns was the smallest. In addition, the displacement vector distribution in the last frame shows that the displacement distribution value of pure water was still the largest among the five systems. With the increase in NaCl concentration, the displacement distribution value of physiological saline droplets gradually decreased, which is because of the increase in NaCl concentration and the overall viscosity of the liquid, which limited the irregular diffusion of molecules in physiological saline to a certain extent.

The local displacement distributions of physiological saline droplets with 0% and 6% NaCl concentration were extracted, as shown in Figure 3. Obviously, the displacement distribution of physiological saline can be divided into three parts: region 1, region 2, and region 3. Interestingly, region 1 and region 3 are distributed on both sides of region 2, and the displacement distribution of the former is much higher than that of the latter, regardless of the concentration of NaCl. However, the concentration of NaCl affects the displacement of molecules in the same region; when the NaCl concentration is small, the displacement distributions of molecules in region 1 and region 3 are higher. Figure 4 shows the infiltration height of physiological saline on the surface of graphene with different NaCl concentrations. The droplet height in Figure 4 is taken from each frame of simulation, and then its average value is calculated to obtain the final infiltration height, which also includes the fluctuation range of each droplet height. As shown in Figure 4, the height of physiological saline droplets increased with the increase in NaCl concentration, and this is because physiological saline cannot be fully dispersed on the surface of graphene under the constraint of surface tension. On the contrary, the fluctuation range of droplet height decreased with the increase in NaCl concentration, which is also because the increase in NaCl concentration leads to an increase in viscosity and an increase in the interaction between molecules in the droplet, and the molecules are bound to each other, resulting in a decrease in displacement.

Compared with the effect of NaCl concentration on the infiltration of physiological saline, the effect of the electric field on physiological saline droplets is more obvious. The infiltration of droplets with 26.35% NaCl concentration was extracted, as shown in Figure 5. With the increase in electric field intensity, the infiltration of droplets on the surface of graphene became more obvious. The infiltration degree of the droplet on the surface of graphene under a 0.02 V/Å electric field intensity was significantly less than that under 0.04 V/Å and 0.06 V/Å electric field intensities. When the electric field intensity reached 0.08 V/Å, as shown in Figure 5c, the droplets were completely flat on the surface of graphene, which is because H_2_O in the droplets and the surface of graphene have polarity and presence of electric field exerts a directional force on it, making H_2_O break the binding of surface tension and have a strong interaction with the surface of graphene [33,34,35].

### 3.2. Concentration Distribution of Physiological Saline

The density distribution of droplets is of great significance to understand the infiltration mechanism of droplets on the surface of graphene. Figure 6 shows the two-dimensional distribution of droplet density distribution in four simulation systems under different NaCl concentrations. As shown in Figure 6, the density distribution of physiological saline droplets at the concentrations of 3% and 6% was relatively uniform. When NaCl concentration reached 13.2%, the internal concentration of droplets close to the surface of graphene was slightly higher, which indicates that when NaCl reaches a certain concentration, NaCl molecules in droplets tend to aggregate on the surface of graphene. When the concentration of NaCl increased to 26%, not only did the local concentration of droplets near the surface of graphene increase, but also the concentration of droplets far away from the surface of graphene increased significantly, which indicates that NaCl accumulated inside the droplets.

In order to explore the effect of electric field intensity on droplet density distribution, the concentration distribution of the simulation system with 26% of NaCl concentration was extracted, and electric field intensities of 0.02 V/Å, 0.04 V/Å, 0.06 V/Å, and 0.08 V/Å were applied. As shown in Figure 7a–c, with the increase in electric field intensity, the infiltration effect of droplets became more and more obvious, which decreased the heights of the droplets, which resulted in an increase of the relative density of NaCl molecular concentration (0.11, 0.15 g/cm^3^, 0.21 g/cm^3^, and 0.24 g/cm^3^, respectively) under higher electric field intensity. Under a high-intensity electric field, due to the decrease in droplet height, the diffusion space of NaCl molecules was sharply compressed, and the NaCl concentration in droplets was higher and more uniform.

### 3.3. Infiltration Angle Distribution of Salty Fluid

In this section, we explore the effects of droplet concentration and electric field intensity on the contact angle of droplets on the surface of graphene. Figure 8 shows the variation in the contact angle of droplets with different NaCl concentrations. It can be seen that when the NaCl concentration increased from 3% to 13.2%, the contact angle of droplets increased from 78.07° to 13.2°, as shown in Figure 8a–c. When NaCl concentration rose to 26%, the contact angle of the droplet jumped from 13.2° to 103°, which was due to the large increase in NaCl concentration in the droplet, resulting in a large change in its viscosity, as shown in Figure 8d. When the droplet volume remained unchanged, the contact surface between the droplet and the graphene surface cannot expand on the graphene surface, resulting in the increase in the contact angle between the droplet and the graphene surface.

Figure 9 shows the effects of different electric field intensities on the droplet contact angle. As shown in Figure 9, the electric field intensity is inversely proportional to the contact angle of droplets. When the electric field intensity was 0.02 V/Å, the droplet contact angle reached the maximum value (97.7°), as shown in Figure 9a. When the electric field intensity increased, the droplet height decreased and the contact angle decreased. When the electric field intensity reached 0.08 V/Å, the contact angle reached the minimum value, i.e., 39.33°, as shown in Figure 9d.

## 4. Conclusions

In this paper, the infiltration of physiological saline droplets on the surface of graphene under different NaCl concentrations and different electric field intensities was studied. The results show that an increase in NaCl concentration reduces the displacement value of molecules in droplets. At the same time, the displacement of the molecules in the contact area on both sides of the droplet is much higher than that in the middle of the droplet. In addition, the increase in NaCl concentration increases the overall concentration of droplets, and the NaCl is easy to aggregate on the surface of graphene. The increase in NaCl concentration makes it more difficult for droplets to infiltrate on the surface of graphene, and the contact angle of droplets increases with the increase in NaCl concentration. The effect of voltage intensity on droplet infiltration is much greater than that of NaCl concentration. With the increase in electric field intensity, the infiltration effect of droplets is more obvious. The droplets were completely tiled on the surface of graphene under an electric field intensity of 0.08 V/Å. The larger the electric field intensity, the smaller the infiltration angle, which is mainly due to the polarity of water molecules and graphene. This study has reference significance for the study of body fluid volatilization on the surface of human skin.

## Figures and Tables

**Figure 1 materials-15-03925-f001:**
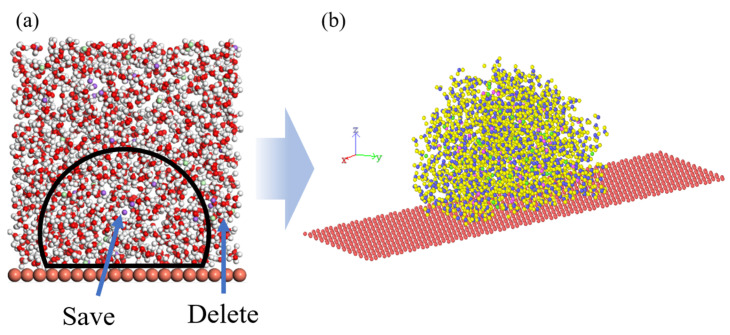
Simulation system of physiological saline. (**a**) The area to be extracted from the initial model, (**b**) the final model for simulation.

**Figure 2 materials-15-03925-f002:**
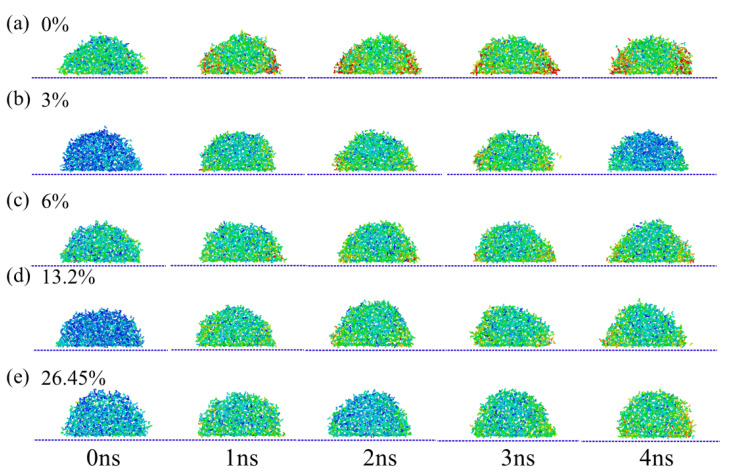
Distribution of molecules under different concentration of NaCl. (**a**) 0%, (**b**) 3%, (**c**) 6%, (**d**) 13.2%, (**e**) 26.45%.

**Figure 3 materials-15-03925-f003:**
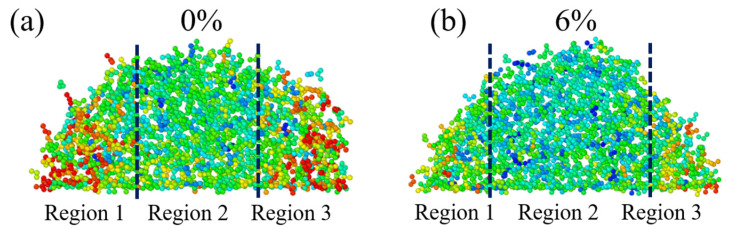
Displacement distribution of physiological saline at different NaCl concentrations, (**a**) 0% NaCl and (**b**) 6% NaCl.

**Figure 4 materials-15-03925-f004:**
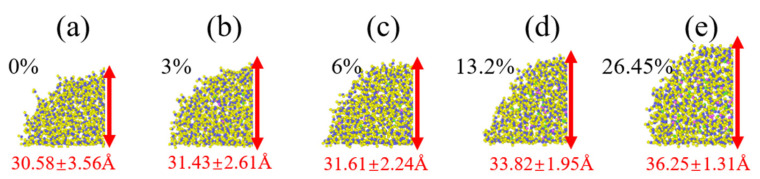
Height of physiological saline at different NaCl concentrations. (**a**) 0%, (**b**) 3%, (**c**) 6%, (**d**) 13.2%, (**e**) 26.45%.

**Figure 5 materials-15-03925-f005:**
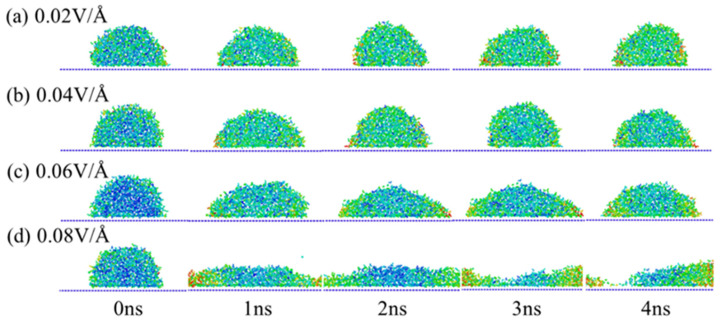
Displacement distribution under different electric field. (**a**) 0.02 V/A, (**b**) 0.04 V/A, (**c**) 0.06 V/A, (**d**) 0.08 V/A.

**Figure 6 materials-15-03925-f006:**
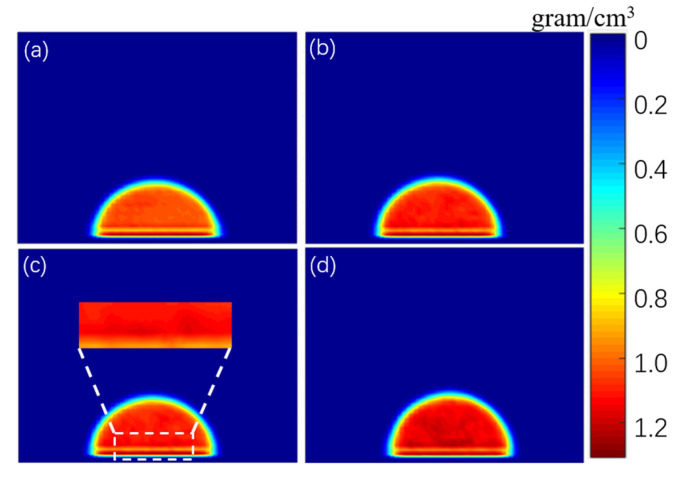
Density distribution of droplets with different NaCl concentrations, (**a**) 3%, (**b**) 6%, (**c**) 13.2%, and (**d**) 26%.

**Figure 7 materials-15-03925-f007:**
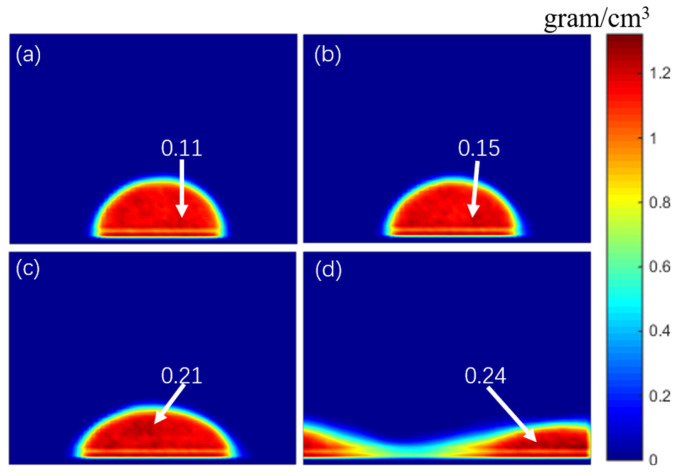
Density distribution of the fluid drop under different electric field intensities, (**a**) 0.02 V/Å, (**b**) 0.04 V/Å, (**c**) 0.06 V/Å, and (**d**) 0.08 V/Å.

**Figure 8 materials-15-03925-f008:**
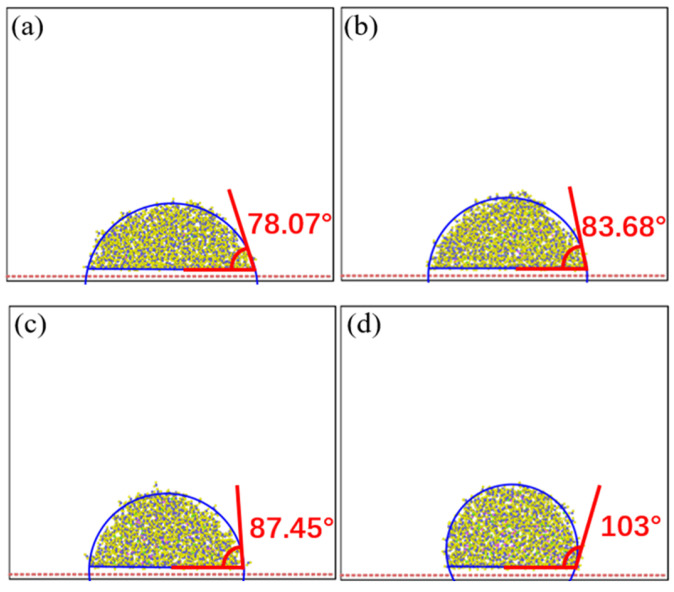
Contact angle of physiological saline under different concentrations of NaCl, (**a**) 3%, (**b**) 6%, (**c**) 13.2%, and (**d**) 26%.

**Figure 9 materials-15-03925-f009:**
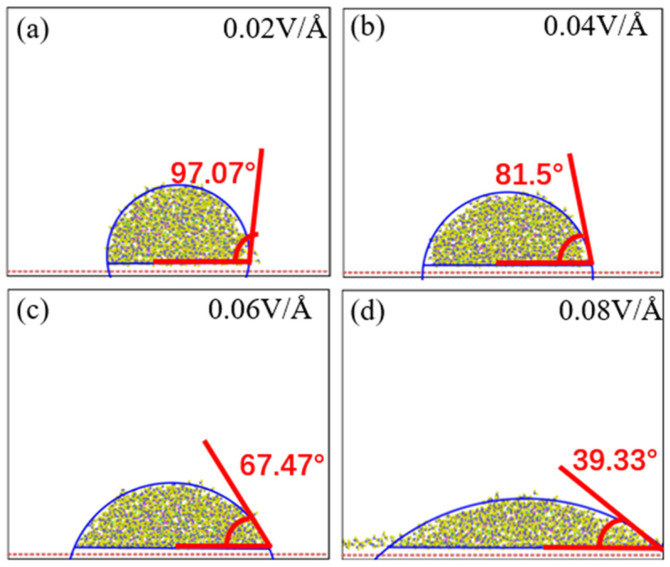
Contact angle of physiological saline under different electric field intensities (**a**) 0.02 V/Å, (**b**) 0.04 V/Å, (**c**) 0.06 V/Å, and (**d**) 0.08 V/Å.

**Table 1 materials-15-03925-t001:** Proportion of NaCl in the system.

System	P-1	P-2	P-3	P-4
ω	3%	6%	13.2%	26%

## Data Availability

The data presented in this study are available on request from the corresponding author.

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
