# Peer review of "Molecular Dynamics Investigation of Spreading Performance of Physiological Saline on Surface"

_materials, 2022, doi:10.3390/ma15113925_

Round 1

Reviewer 1 Report

The assessed manuscript describes some results on inorganic salts on the surface of graphene. The manuscript contains a significant number of typos and language inconsistencies. The manuscript does not provide comparison with the available experimental results and prior computer simulations. 

The manuscript can only be reconsidered if it is properly revised. The authors shall provide an adequate introduction that argues the necessity to carry out such a study and place their work in the context of international research. There are numerous flaws in this manuscript that must be removed.

The manuscript contains typos including typos in chemical terms. In general, the level of writing is low. The text contains numerous errors that could have been fixed by simply using automatic grammar checking. Professional proofreading is necessary.

The introduction does not provide any goal setting and any discussion of the competitive works.

The reference list contains plenty of irrelevant citations. Many of them do not suit this work even by their titles.

Fig. 1. The simulated systems shall be visualized.

Fig. 2: caption shall be freed of grammar errors.

Fig. 3. The caption is not descriptive of the figure content.

Fig. 4. Error bars must be provided. Significant figures must be observed. Tab © reports three digits whereas other tabs report two digits.

Fig. 8, fig. 9: terminology must be correct in English.

Comparison shall be provided with the published results of ions on the graphene surface: "The band gap of graphene is efficiently tuned by monovalent ions","Can inorganic salts tune electronic properties of graphene quantum dots?", "Exfoliation of graphene in ionic liquids: pyridinium versus pyrrolidinium", "Graphene/ionic liquid ultracapacitors: does ionic size correlate with energy storage performance?", "Graphene exfoliation in ionic liquids: unified methodology"

The effect of charge, polarization, structure, electrostatic versus London forces shall be compared and discussed in a clear and  scientific manner. The role of solvent must be discussed as an important factor of the studied process.

The following sentence does not not make any sense: 'MD simulation of evaporation process does not need some assump- 44

tions of computational fluid dynamics [20-21]."

The following sentence is physically illiterate: "Concentration distribution of liquid drop under different concentration of NaCl ions."

Fig. 7: unsatisfactory quality. The data is unreadable.

Author Response

We have uploaded a file for your questions, please see the attached file.

Reviewer 2 Report

The manuscript explores the spreading performance of physiological saline droplets on the surface of graphene under different NaCl concentration and electric field intensity.
The results show that the increase of NaCl concentration reduce the displacement vector value of molecules in droplets. In addition, NaCl is easy to aggregate on the surface of graphene.
The experimental results are trustworthy and fitted into the current state of the carbon nanoscience concepts. 
I favor such a kind of works but ought to ask for a minor revision dealing with connecting the work with other relevant papers and clearer representation of the results.

A few notes:

1) The reason for choosing the proportion of NaCl in the system is not clear.

2) It is not obvious to use a graphene surface as a simplified skin model.

3) In this work, the authors consider ideal graphene, but do not take into account curvature, impurities, and structural defects, which can significantly affect the surface wettability.

Author Response

(The authors gave the same response as above.)

Round 2

Reviewer 1 Report

Happy researching!